# Systemic Immune Modulation Induced by Ephedrine in Obese-Diabetes (*db/db*) Mice

**Seung-hoon Lee** [1] , **Hyunah Lee** [2,*] **and Rackhyun Park** [1,*]

1    Department of Life Science, Yongin University, 470 Samga Dong, Cheo-In Gu,
     Yong-In Si 17092, Republic of Korea; shlee@yongin.ac.kr
2    Immunecell Therapy Research Center, Seoul Song Do Colorectal Hospital, 78 Dasan-ro, Jung-gu,
     Seoul 04597, Republic of Korea
*    Correspondence: halee61@gmail.com (H.L.); flowblue@yongin.ac.kr (R.P.); Tel.: +82-31-8020-2778 (R.P.)

**Abstract:** Immune-modulatory effects in obese-diabetes (*db/db*) mice were observed to understand the possible mechanism(s) of ephedrine-induced unfavorable responses. The ephedrine doses were selected based on the FDA report (NTP Tech Rep Ser NO 307; CAS# 134-72-5), which showed the non-toxic dose for B6C3F1 mice. In *db/db* mice, higher doses (6 and 12 mg/mouse) of ephedrine significantly harmed the liver and lung morphology, including fatty liver with multiple blood vessel engorgement, alveolar wall thickening, and inflammatory response in the lung. The immune micro-environment of *db/db* mice was an inflammatory state with suppressed adaptive cellular immunity. After the administration of ephedrine, significant deterioration of NK activity was observed with lowered gene transcription of klrk1 encoding NKG2D, and of ccl8, a NK cell targeting chemokine. Suppressed cellular immunity in *db/db* mice was lowered ever further by single ephedrine treatment, as was evidenced by mitogen-induced T or B cell proliferations. These observations demonstrate that at the non-toxic doses in normal B6C3F1 mice, ephedrine clearly suppressed systemic immunity of *db/db* mice. The data suggest that the immune micro-environment of obese individuals is fragile and susceptible to ephedrine-related pathologic response, and this may be a prelude to the induction of obesity-related secondary immunological disorders.

**Keywords:** ephedrine; obesity; immunity; obese-diabetes(*db/db*)

## 1. Introduction

Obesity-related incurable health problems are a social issue that has therapeutic concerns. Pharmacological therapeutics for obesity, including sympathomimetic drugs, are classified based on their mechanism of action: suppressors of appetite or fat absorption and stimulators of energy expenditure or thermogenesis [1–3]. A sympathomimetic agent, ephedrine enhances the sympathetic neuronal release of norepinephrine and epinephrine, which makes it useful for bronchial airway expansion, suppression of nasal bleeding or blood pressure induction. As an anti-obesity drug, ephedrine may directly stimulate brown adipocyte respiration. However, inhibition of appetite, increasing energy consumption, and weight loss are mainly due to the β-adrenergic receptor stimulation [4,5]. Adrenergic receptor stimulation also induces cardiovascular side effects, which are a major limiting factor in the use of ephedrine in obese individuals. In the US, the use of ephedrine has been banned [6,7]. However, in other countries, including Asian countries, ephedrine in the "Ma Hwang" is being used in herbal medicine as an anti-obesity agent.

The link between the obesity-related pathology and immune modulation has been demonstrated in an animal model as well as in human patients. Although ephedrine is useful for short-term weight loss, it may affect systemic immunity in obese individuals as a pathological mechanism. The ephedrine-induced alteration of β-adrenergic receptors in the human lymphocytes was observed [8,9]. In NZM391 mice, beta-adrenergic receptor activation by ephedrine could cause immune modulation, resulting in aggravation of sys-

temic lupus erythematous [10]. But until now, ephedrine-induced modulation of systemic immunity has not been fully observed.

In patients, the obesity phenomenon is characterized as low-grade systemic inflammatory response syndrome (SIRS), which is similar to sepsis induced by gram-negative bacteria [11–14]. Obesity-related cytokine dysregulation and lipotoxicity can cause insulin resistance and metabolic disease, which can increase the lethality by causing internal organ destruction or infection [12,13]. Due to the immune-inflammatory micro-environment, obese individuals may be more susceptible to the effects of ephedrine on the immune system, which may be a prelude to the induction of secondary diseases like cancer, diabetes, or rheumatoid arthritis.

Our previous data reveal significantly elevated immune-inflammatory status but suppressed adaptive cellular immunity in genetically modified obese-diabetes mice (leptin receptor knock-out *db/db* mice) [15]. The data confirm increased susceptibility of obese diabetic mice to alcohol-induced pathologic situations with suppressed adaptive immunity. Thus, the present demonstration of ephedrine-induced alteration of systemic immunity in the obese diabetic mouse model (*db/db* mice) may expand our understanding of obesity. These results suggest that ephedrine can be considered to accelerate toxicity through the immunomodulatory effects in obese conditions.

## 2. Materials and Methods

### 2.1. Animals

Specific pathogen-free female *db/db* mice as an obese-diabetic animal model and C57BL/6J mice, a background control animal of 4~5 weeks of age, were obtained from Charles River Japan, Inc. (Yokohama, Japan). The *db/db* mouse was delivered from the background strain C57BL/6J by spontaneous mutation of leptin receptor (a/a +Leprdb/ + Leprdb). Without diet control, the *db/db* mouse became obese at around 3~4 weeks of age with elevation of plasma insulin level beginning at 10~14 days of age and of plasma sugar level at 4~8 weeks of age (strain information from The Jackson Laboratory, Bar Harbor, ME, USA). Five- or six-week-old mice were used in this study. Specific pathogen-free C57BL/6J naïve mice and *db/db* mice were provided with water and food ad libitum and quarantined under 12 h of light. There was also a 12 h dark photoperiod in the animal care facility. Animal care was performed following the Institute of Laboratory Animal Resources (ILAR) guidelines. The mice were acclimated for at least one week before any experiments were conducted.

### 2.2. Chemicals

RPMI-1640 medium, fetal bovine serum, and penicillin-streptomycin were obtained from GIBCO laboratories (Grand Island, NY, USA); Con A (Concanavalin A, from *Canavalia ensiformis*), LPS (Lipopolysaccharides, from *Escherichia coli* 055:B5), and Dextran-FITC were obtained from SIGMA Chemical Co. (St. Louis, MO, USA). [3H]-Thymidine and Na251CrO4 were purchased from Perkin Elmer Corp. (Norwalk, CT, USA) Antibodies for phenotype analysis were obtained from eBioscience (San Diego, CA, USA).

### 2.3. Cell Line

A target of natural killer (NK) cell activity, YAC-1 mouse lymphoma cell line was purchased from the American Type Culture Collection (ATCC). (Rockville, MD, USA) Cell line was maintained in RPMI-1640 medium supplemented with 10% heat-inactivated fetal bovine serum (FBS), 2 mM glutamine, 100 U/mL penicillin, and 100 µg/mL streptomycin (complete medium).

### 2.4. Ephedrine Treatment Schedule

An experimental dose of ephedrine was determined based on the FDA report titled "Toxicology and Carcinogenesis Studies of Ephedrine Sulfate in F344/N Rats and B6C3F1 Mice" (NTP Tech Rep Ser NO 307; CAS# 134-72-5). The doses of ephedrine selected for this study were proven to be non-toxic in B6C3F1 mice by observing weight loss, morphological changes, and survival. Ephedrine (ephedrine hydrochloride, Sigma-Aldrich, WI, USA) was dissolved in saline and orally administered using a zonde needle into the *db/db* and C57BL/6 mice. Each mouse was treated with ephedrine at doses of 3, 6, and 12 mg/mouse (120 ~ 500 mg/kg in the FDA report CAS# 134-72-5) in the single treatment schedule. In the preliminary experiment, more than 12 mg of single ephedrine administration resulted in the death of *db/db* mice in 2 h. Therefore, the highest single dose was set at 12 mg. Control mice were orally treated with saline. Twenty-four hours after the administration of ephedrine, the mice were sacrificed by cervical dislocation to analyze the systemic immunity using splenocytes and plasma. Other organs, including lung, liver, and kidney, were eliminated, paraffin-embedded, and stained with H&E to observe the histo-pathological alterations. Histological observation was performed under the microscope (Leica DM 3000). The picture was presented as ×200 magnification.

### 2.5. Phenotype Analysis by Flow Cytometry

Surface markers of immune cells from splenic lymphocytes were determined. Single cells from the C57BL/6J or *db/db* mice splenocytes ($1 \times 10^6$ cells/mL) with or without ephedrine treatment were incubated with fluorescence (FITC or PE)-labeled surface antibodies in PBS with 0.1% sodium azide and 1% FBS (PBS-CS) for 40 min at 4 °C. Either FITC or PE-conjugated anti-mouse anti-CD19 (for B cells), CD3, CD4, CD8, CD25 (for T cell subsets), CD11b, Mac3, F4/80 (for macrophages), DX-5 (for NK cells), and CD11c (for dendritic cells) were selected to analyze the alterations. Within 2 h of antibody labeling, the cells were analyzed in the flow cytometer (FACS Vantage, Becton-Dickinson, Mountain View, CA, USA).

### 2.6. Immune Cell Proliferation Assay

To define the function of immune cells from splenic lymphocytes, induction of proliferation by mitogens Con A (for T cells) or LPS (for B cells) was observed by 3H-Thymidine incorporation assay using the method performed in the author's previous study [15]. Responder splenic lymphocytes ($1 \times 10^6$ cells/mL) from C57BL/6 or *db/db* mice with or without ephedrine treatment were incubated with a 1 μg/mL final concentration of either Con A or LPS for 96 hrs. Eighteen hours before the harvest, 1μCi 3H-Thymidine was pulsed. Cells were harvested on the glass fiber filter (Whatman, Maidstone, England) using a PhD cell harvester (Cambridge Technology, Inc., Cambridge, MA, USA). Scintillation cocktail (Beckman, Fullerton, CA, USA) was added onto the filter, and the radioactivity incorporated into the proliferating cells was measured by a β-scintillation counter (Beckman LS6500, Fullerton, CA, USA).

### 2.7. NK Activity

NK activity was measured by 51Cr–release assay from the Na251CrO4 labeled YAC-1 ($1 \times 10^5$ cells/mL) target cells (T) co-incubated with effector splenic lymphocytes (E) from the ephedrine-treated mice for 4 h. Radioactivity in the supernatants separated from the incubated cells was measured by a Wallac 1470 Wizard gamma counter (Finland). Spontaneous release (SR) and total release (TR) were measured in the supernatants of target cells incubated with either the culture medium or 1N HCl. Percent NK-cell activity was calculated as follows:

%NK cell activity = [(Experimental cpm − SR cpm) ÷ (TR cpm − SR cpm)] × 100

## 2.8. Macrophage Phagocytosis

Phagocytic ability of F4/80+ macrophages was determined by measuring Dextran-FITC uptake with a flow cytometer (FACS Vantage, Becton-Dickinson, Mountain View, CA, USA). Splenic lymphocytes were labeled with anti-F4/80-PE antibodies and incubated with Dextran-FITC. Double positive (F4/80-PE+ and Dextran-FITC+) cells were analyzed as phagocytic macrophages.

## 2.9. Cytokine Micro-Bead Assay

Cytokines secreted into the blood were measured by micro-bead assay (set of assaying IFN-$\gamma$, IL-2, IL-12p70, IL-12p40, IL-10, TNF-$\alpha$, VEGF) using plasma. Mouse blood was obtained by ocular-venous puncture using a non-heparinized capillary tube and immediately centrifuged at 3000 rpm for 10 min to separate the plasma, then aliquoted and stored at $-70\ ^\circ$C until assayed. Multiple cytokine analysis kit was obtained from Upstate Cell Signaling Solutions (Lake Placid, NY, USA). Millipore multi-screen 96 well filter plates (Bedford, MA, USA) were used in multiplex cytokine kits. Assays were run in triplicate according to the manufacturers' protocol. Data were collected using the Luminex-100 system Version 1.7 (Luminex, Austin, TX, USA). Data analysis was performed using the MasterPlex QT 1.0 system (MiraiBio, Alameda, CA, USA).

## 2.10. Gene Microarray: Target Labeling and Hybridization of Microarray

cDNA microarray was performed with lymphocytes separated from the spleen. For DNA microarray experiments, the synthesis of target cRNA probes was performed using Agilent Low RNA Input Linear Amplification kit (Agilent Technology, Santa Clara, CA, USA) according to the manufacturer's instructions. Briefly, 1 μg of the total RNA was labeled with Cy3 or Cy5 fluorescent dye using Agilent labeling kit. Labeled cRNA was purified on RNase mini column (Qiagen) according to the manufacturer's protocol. Labeled cRNA target was quantified using ND-1000 spectrophotometer (NanoDrop Technologies, Inc., Wilmington, DE, USA).

## 2.11. Statistical Analysis

Each experimental group contained 5~6 mice for one experiment. The experiments were repeated 3~4 times using the same protocol. Data are expressed as mean $\pm$ standard error (SE). Significant differences among the treatment groups were calculated by analyses of variance (ANOVA) using Bonferroni correction. Differences were considered to be statistically significant at $p < 0.05$

## 3. Results

### 3.1. Effects of Ephedrine on the Tissues and Body Weight

At the beginning of the experiments, the average body weight of *db/db* mice was about twice greater than that of normal C57BL/6J mice (41.53 + 1.24 g vs. 22.5 + 0.76 g for *db/db* mice and C57BL/6J mice, respectively). Severe internal organ atrophy was observed with packed visceral fat in *db/db* mice. Single administration of ephedrine did not significantly affect the body weight of C57BL/6J or *db/db* mice. The most significant pathologic finding induced by ephedrine administration in the liver was multiple blood engorgement, which was observed only in the *db/db* mice (Figure 1A). Unlike liver toxicity induced by higher doses of ephedrine, diffused thickening of the alveolar wall was observed at all doses in the *db/db* mice (Figure 1B).

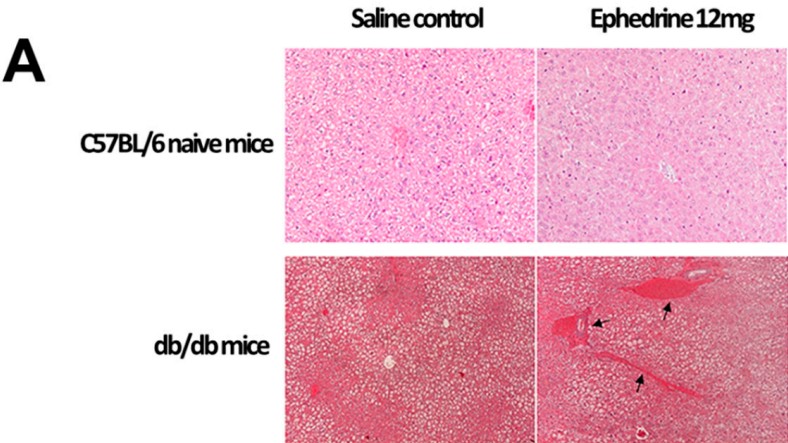

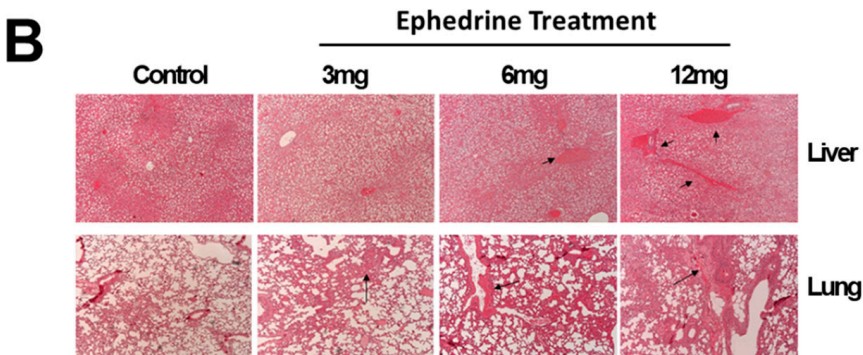

**Figure 1.** Histo-pathological changes induced by ephedrine administration. (**A**) Liver of normal C57BL/6 and *db/db* mouse. (**B**) Lung and liver of *db/db* mouse. Sites for the blood engorgement (liver) or alveolar wall thickening and inflammatory response (lung) were indicated with arrows. H&E stained tissues were observed under the microscope with ×200 magnification.

### *3.2. Ephedrine Reduced the Number of Splenocytes in db/db Mice*

With significant spleen tissue atrophy, the total number of splenocytes was lower in *db/db* mice compared to C57BL/6J mice. Administration of ephedrine significantly reduced the number of splenocytes in *db/db* mice (Figure 2A). On the other hand, the tendency to induction of splenocyte number was observed in C57BL/6 mice by lower doses of ephedrine treatment (Figure 2A). The number of CD3+ T cells and CD19+ B cells was lower in *db/db* mice than in C57BL/6J mice (Figure 2B). Ephedrine tended to reduce both the T and B cell number in *db/db* mice, but not in C57BL/6J mice (Figure 2B). The same effect of ephedrine was observed for T cell subtype alteration. After ephedrine treatment, the number of CD4+ helper T cell and CD8+ cytotoxic T cell subtype was slightly induced in C57BL/6J mice but reduced in *db/db* mice (Figure 2C). The number of immune regulatory cell was not affected significantly by single administration of ephedrine, as was shown with a CD4+ 25+ regulatory T cell and Gr1 + CD115+ myeloid-derived (monocytic) suppressor cell population in both strains of mice (Figure 2D). The number of dendritic cells (CD11c+) or NK cells (DX5+) tended to decrease by ephedrine in *db/db* mice (Figure 2E). Monocyte (CD11b+) number was increased by ephedrine in C57BL/6J mice but decreased in *db/db* mice (Figure 2F). Modulation of the activated macrophage (Mac3+) number was significant in *db/db* mice but not in C57BL/6J mice (Figure 2F).

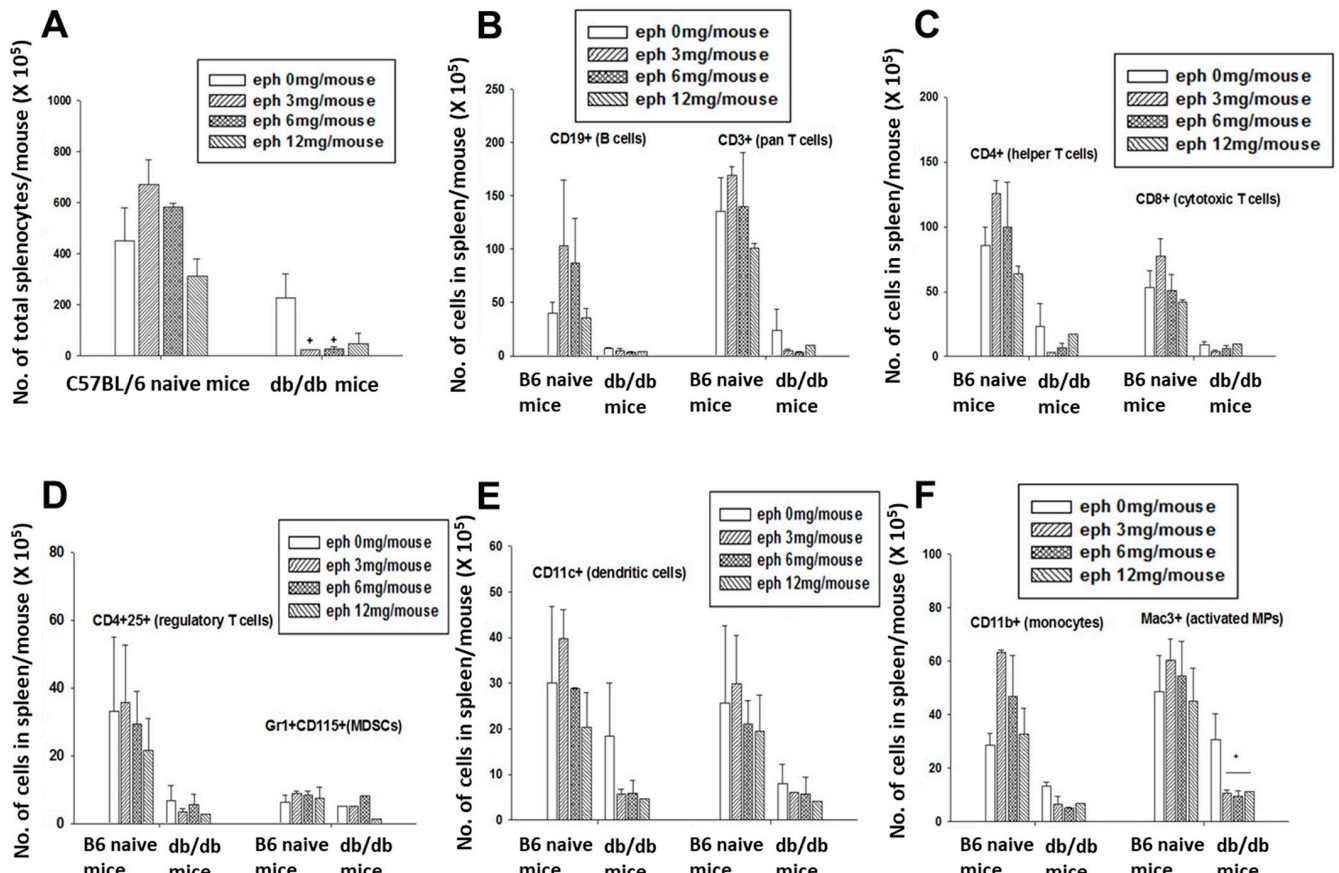

**Figure 2.** Effect of ephedrine on the splenic immune cell number. Gross number of splenocytes expressing the following phenotypes was counted. (**A**) Total splenocytes. (**B**) CD3$^+$ T cells and CD19$^+$ B cells in the spleen. (**C**) T cell subtypes: CD4$^+$ helper T cells, CD8$^+$ cytotoxic T cells. (**D**) Immune regulatory (suppressor) cells: CD4$^+$25$^+$ regulatory T cells, Gr1+CD115+ myeloid-derived suppressor cells. (**E**) CD11c$^+$ dendritic cells and DX5$^+$ natural killer cells. (**F**) CD11b$^+$ monocytes and Mac3$^+$ activated macrophages. Plus signs represent the statistical significance of ephedrine effect in the *db/db* mice (+ $p < 0.05$).

### 3.3. Effects of Ephedrine on the Natural Killer Cell (NK) Activity

Without ephedrine administration, the background level of NK cell activity in *db/db* mice was significantly higher than that in normal C57BL/6J mice, which confirmed the immune-inflammatory microenvironment in the *db/db* mice (Figure 3). Splenic NK cell activity of *db/db* mice was significantly repressed below the detection limit by administration of ephedrine (Figure 3). On the other hand, ephedrine treatment increased the NK activity in C57BL/6J mice (Figure 3).

### 3.4. Effects of Ephedrine on the Macrophage Function

Ephedrine administration affected the splenic-macrophage phagocytic function. Among the splenocytes, dextran uptaking F4/80 + cells (F4/80-PE + dextran-FITC+ double positive detected by flow cytometry) were considered as phagocytic macrophages. Without external stimulation, phagocytic macrophage proportion was similar in *db/db* mice and in C57BL/6J mice (Figure 4). However, induction of phagocytic macrophage response to ephedrine administration (lower doses; 3 or 6 mg/mouse) was significant only in *db/db* mice (Figure 4).

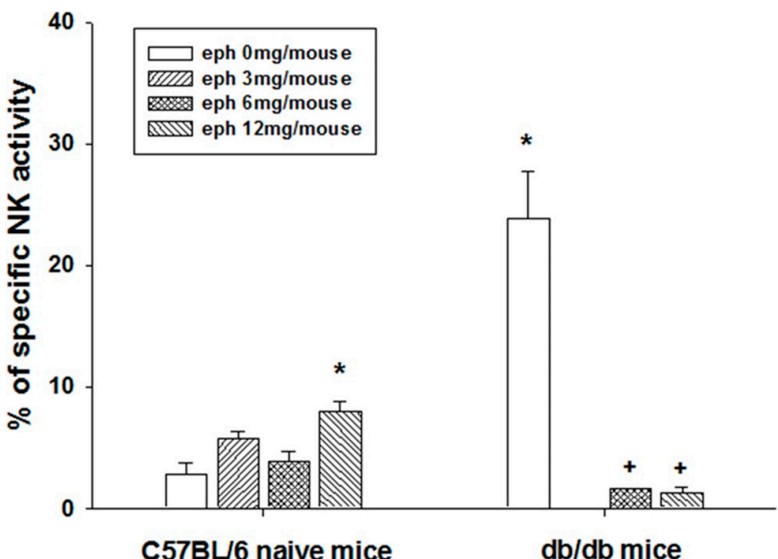

**Figure 3.** Effect of ephedrine on the splenic NK activity. NK activity was measured after single administration of ephedrine by $^{51}$Cr release assay from the $Na_2^{51}CrO_4$ labeled YAC-1 ($1 \times 10^5$/mL) target cells (T) incubated with effector splenic lymphocytes (E). Asterisk represents the statistically significant difference compared to C57BL/6 mice control (* $p < 0.05$). Plus signs indicate the statistical significance induced by ephedrine administration in the *db/db* mice (+ $p < 0.05$).

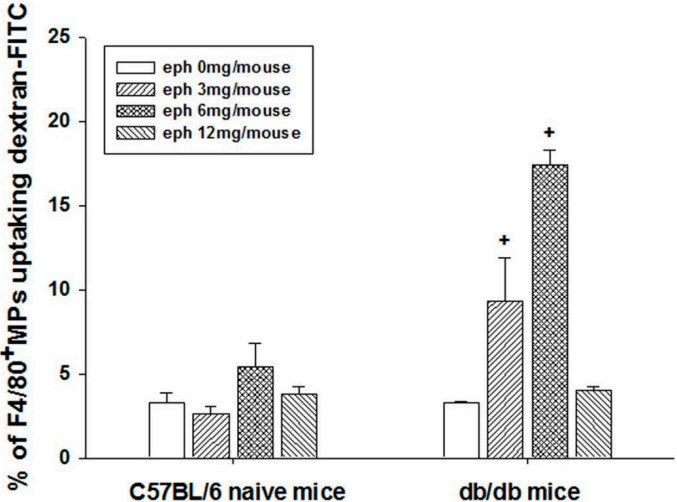

**Figure 4.** Effect of ephedrine on the macrophage (MP) phagocytosis. Phagocytic ability was determined by measuring dextran-FITC taken F4/80$^+$ MPs by flow cytometry. F4/80-PE and Dextran-FITC double positive cells were regarded as phagocytic MPs. Plus signs indicate the statistical significance induced by ephedrine administration in the *db/db* mice (+ $p < 0.05$).

### 3.5. Effects of Ephedrine on Mitogen-Induced Lymphocyte Proliferation

Compared to normal C57BL/6J mice, splenic lymphocyte function was significantly defective in the *db/db* mice as determined by a mitogen-induced lymphocyte proliferation assay (Figure 5). With ephedrine administration, the background proliferation level and the responses to the T cell-specific mitogen, concanavalin A (ConA), and the B cell-specific mitogen of bacterial origin, lipopolysaccharide (LPS), were significantly inhibited in *db/db* mice (Figure 5). In C57BL/6J mice, the splenic lymphocyte proliferation without mitogen stimulation was not altered by ephedrine administration. However, the mitogen-induced T or B cell proliferation was inhibited by single administration of ephedrine also in C57BL/6J mice (Figure 5).

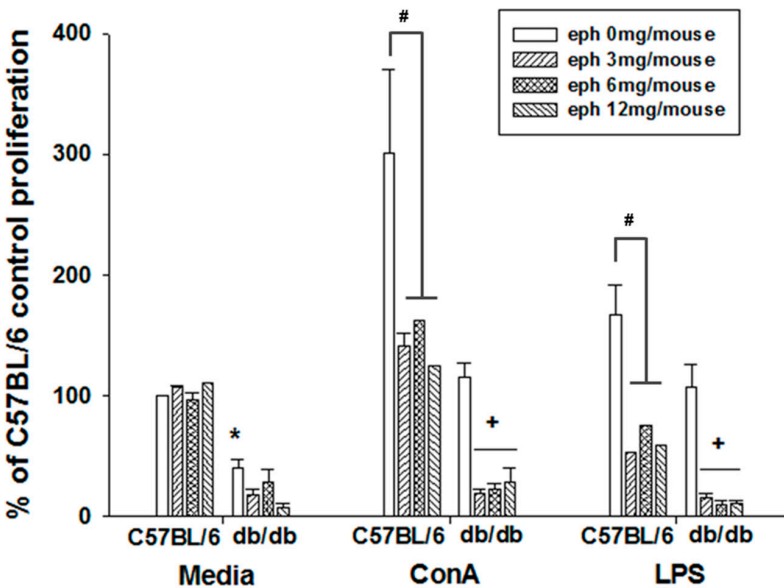

**Figure 5.** Effect of ephedrine on the mitogen-stimulated lymphocyte proliferation Either ConA, a T cell mitogen, or LPS, a B cell mitogen, induced splenic lymphocyte proliferation that was analyzed by $^3$H-incoporation into the DNA of proliferating cells. Ephedrine effects were plotted as the relative change from the saline-administered C57BL/6J splenocytes' proliferation without external mitogen stimulation. Asterisks indicate the significant difference between the *db/db* mice and C57BL/6 mice (* $p < 0.05$). Plus signs represent the significant (+ $p < 0.05$) effect of ephedrine in the *db/db* mice. Hash signs represent the significant ($^\#$ $p < 0.05$) effect of ephedrine in the C57BL/6 mice.

### 3.6. Alteration of Cytokine Secretion Induced by Ephedrine

Plasma levels of IFN-γ, IL-2, IL-12p70, IL-12p40, IL-10, and TNF-α, and vascular endothelial growth factor (VEGF) were determined in the cytokine detection set. The plasma levels of IFN-γ, IL-2, IL-12p40, TNF-α, and VEGF were under the detection limit (<10 pg/mL) and/or were not altered by ephedrine administration in *db/db* mice. The levels of IL-12p70 and IL-10, the representative cytokines for modulation of T cell-related adaptive immunity, were detectable and significantly lower in *db/db* mice compared to C57BL/6J mice (Figure 6A,B). Single administration of ephedrine increased the plasma level of IL-12p70, a Th1 response inducer, in *db/db* mice but decreased the level in C57BL/6J mice without dose-dependency (Figure 6A). In C57BL/6J mice, the plasma level of IL-10 was significantly decreased with lower doses (3 or 6 mg/mouse) ephedrine (Figure 6B). However, ephedrine-induced alteration was not significant for the IL-10 secretion in *db/db* mice (Figure 6B).

### 3.7. Effect of Ephedrine on Immune-Related Gene Expression (Microarray Analysis)

To observe the differential gene expression related to the ephedrine-induced immune modulation, RNA was prepared from the spleens of *db/db* mice and microarray analysis was performed. Single administration of ephedrine lowered the transcription levels of the antigen-specific immunity-related genes in *db/db* mice. Lowered levels of genes encoding major histocompatibility complexes (H2-Ob, H2-Q1), co-stimulatory molecule (Cd86), and heat shock proteins (Hspa12a, Hspa1a for HSP70) were observed (Figure 7). Reduced levels of klrk1 (killer cell lectin like receptor subfamily k) encoding NKG2-D and ccl8 encoding chemokine targeting NK cells were observed, suggesting the mechanism(s) significantly decreased NK activity in the ephedrine treatment group. Genes for CXCL4 or 7, the chemokines that play a role in angiogenesis-angiostasis, were induced along with the calcium channel-related genes like calm3 (calmodulin 3) and cacnb1. The gene for a fatty acid translocase, CD36, was induced by ephedrine in *db/db* mice.

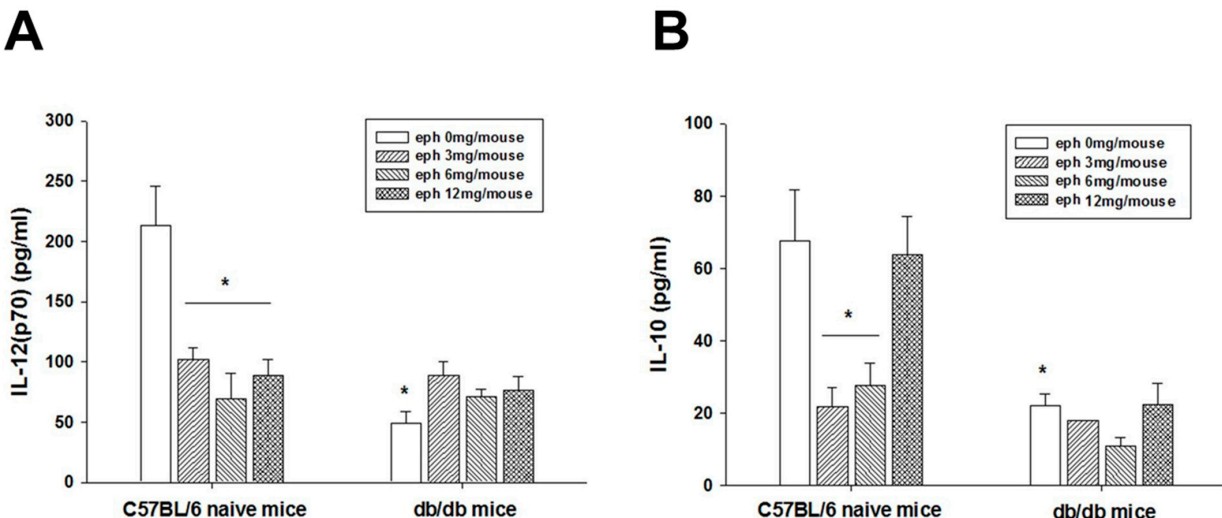

**Figure 6.** Effect of ephedrine on the plasma cytokine level. Ephedrine-induced cytokine secretion ((**A**) IL-12 and (**B**) IL-10) into the blood was observed. Asterisks indicate the significant difference compared to C57BL/6 mice control administered with saline (* $p < 0.05$).

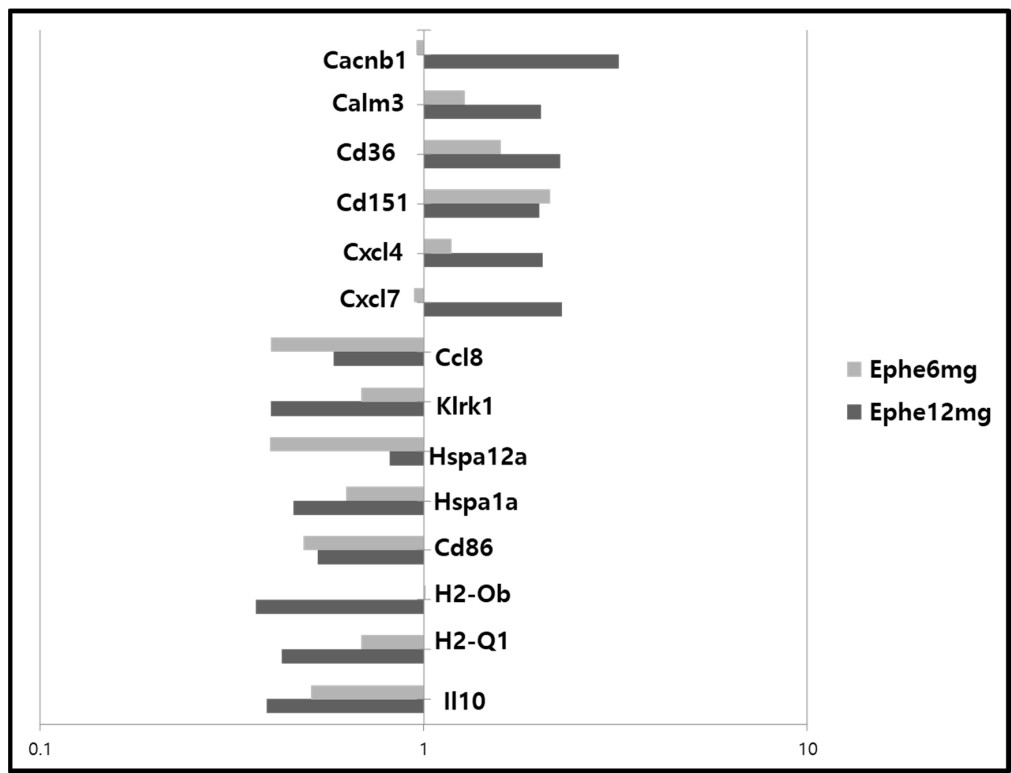

**Figure 7.** Effect of ephedrine on the immune-related gene expression (microarray analysis). Transcriptional alteration of immune-related genes by single administration of ephedrine was observed in the splenic lymphocytes from the *db/db* mice. Data represented as saline-treated control-based changes. *x*-axes presented log-scale changes. *y*-axes indicated the name of altered genes.

## 4. Discussion

Our findings show that ephedrine suppressed systemic immunity and the immune micro-environment; this was particularly observed in *db/db* mice.

Obesity is a chronic immune-inflammatory disease that causes incurable adult health problems such as type II diabetes, cardiovascular disorders, and cancer [16–19]. Previously,

we reported the baseline systemic immune-inflammatory status of *db/db* mice, which was suppressed in adaptive cellular immunity [15]. Our results showed that without the effects on the body weight, single treatment of ephedrine significantly harmed the liver and lung morphology of *db/db* mice but not of C57BL/6J background wild type mice (Figure 1). The fatty liver of *db/db* mice worsened with multiple blood vessel engorgement after ephedrine administration. Also, the alveolar wall thickening and inflammatory response in the lung of *db/db* mice were the pathologic phenomena observed with ephedrine treatment. Both the innate and adaptive immunity of *db/db* mice was modulated by ephedrine treatment. In the *db/db* mice, macrophages and natural killer cells of the innate immune-inflammatory system were significantly activated without any external stimuli, comparable to infection-induced immune responses in naïve mice [20,21]. This abnormal immune-inflammatory status of the *db/db* mice was altered after the administration of ephedrine. Significant deterioration of NK activity was observed after the ephedrine administration (Figure 3), with lowered transcription of klrk1 (Figure 7) encoding NKG2D, the killer cell activating receptor. Also, a reduced level of ccl8 transcription, a gene for NK cell targeting chemokine was observed (Figure 7). These may represent the molecular mechanisms of ephedrine-induced NK activity inhibition in *db/db* mice. On the other hand, the macrophage function tended to increase without statistical significance with ephedrine treatment, although the cell number in the spleen was reduced. Decreased proliferative function of splenic lymphocytes, including both T and B cells in *db/db* mice, was lowered even further by ephedrine, indicating the suppression of adaptive immunity. With the inhibited immune cell proliferative responses, the reduction of genes for the antigen-recognition for adaptive cellular immunity induction was observed in the ephedrine-treated *db/db* mice. Major histocompatibility (MHC) molecules (H2-Ob, H2-Q1), co-stimulatory molecule Cd86, and heat shock protein Hsp70 (Hspa12a, Hspa1a) gene expression decreased by ephedrine (Figure 7). Generally, optimal activation of T cells requires two signals. The first signal is generated by binding of the T cell receptor (TCR) to Ag-MHC complexes on the Ag-presenting cell (APC). The second signal involves interaction of CD80 and CD86 on an APC with CD28 that is expressed on T cells [10,12,22]. Resting B cells do not express CD86, and interaction of CD86 with CD28 delivers positive signals for T cell and B cell activation [11,23,24]. These results suggest that ephedrine-inhibited CD80, H2-Ob, and H2-Q1 expression may cause inhibition of B cell and T cell activation and proliferation and induce impairment of antigen-specific cellular immunity [25–27].

Induction of cxcl4 and cxcl7, the genes for angiostasis–angiogenesis-related chemokines, was observed after the ephedrine treatment in obese-diabetes mice [28–31]. These alterations may have a pathological significance in the ephedrine-induced cardiovascular effects through the alteration of angiogenesis, but they need to be studied further.

In this study, the correlation between the already known adrenergic receptor-related activity and systemic immune modulation was not investigated. Further studies exploring the relationship between these two mechanisms of action would be valuable.

Although this study reveals the immune-modulatory phenomena induced by ephedrine without detailed mechanism discussion, ephedrine clearly alters the systemic immunity of *db/db* mice, suggesting the possible immuno-pathological effect of ephedrine on obese individuals. These observations suggest that ephedrine treatment induces changes in systemic immunity, notably exacerbating the situation in mice with compromised immune systems, such as *db/db* mice. In summary, this study shows that ephedrine-induced deterioration of NK activity and cellular immunity may increase the susceptibility of obese individuals to infection and cancer.

**Author Contributions:** Conceptualization, H.L. and S.-h.L.; methodology, S.-h.L. and R.P.; investigation, H.L. and S.-h.L. writing—original draft preparation, H.L. and S.-h.L., writing—review and editing, H.L. and R.P. All authors have read and agreed to the published version of the manuscript.

**Funding:** This research was funded by National Research Foundation of Korea (NRF), grant number '2020R1F1A107633712' and 'RS-2023-00239974'.

**Institutional Review Board Statement:** The animal study protocol was approved by the Institutional Review Board (or Ethics Committee) of YONGIN UNIVERSITY (YUACUC-2023-06).

**Informed Consent Statement:** Not applicable.

**Data Availability Statement:** Data are contained within the article.

**Conflicts of Interest:** The authors declare no conflict of interest.

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
