# Peer review of "Systemic Immune Modulation Induced by Ephedrine in Obese-Diabetes (db/db) Mice"

_cimb, doi:10.3390/cimb45120630_

Round 1

Reviewer 1 Report

Comments and Suggestions for Authors

 [The significance of the research]

Obesity causes systemic inflammatory responses that further cause several diseases such as diabetes, and cancers. To treat with it, Ma Hwang, which is a traditional medicine in Asian countries, has been used. Its central ingredient ephedrine is a beta-androgenic receptor agonist. It was also used for obesity but was banned by the FDA because of its severe side effects. The authors tried to reveal the underlying mechanisms of the side effects by comparing the differences between normal and obesity model mice, in this study. This study has significance in the toxicology in the treatment of obesity.

 [Title and Purpose]

The authors declare the purpose of this manuscript in the ABSTRACT section that “to define the possible mechanism(s) of ephedrine-induced unfavorable responses as an anti-obesity drug” , but in the introduction, “the present demonstration ~~ may expand our understanding of obesity and related diseases in terms of the mechanism of induction or development of therapeutics”. Also, in the concluding remarks of the Abstract say “this may be a prelude to the induction of obesity-related secondary immunological disorders” but that of the Discussion says “The present demonstration ~~ expand the our understanding obesity and related disease ~~”.

In the manuscript, the authors change the goal from “to reveal the effects (and mechanisms) of ephedrine on obesity” to “to understand the pathology of obesity itself (by using the ephedrine)”. Please clear that point.

For this point, the reviewer will reject this manuscript once but recommend re-submit after reconstruction of the story because the whole data that the authors collected was well demonstrated.

 [Logics]

The authors first demonstrate ephedrine causes multiple blood engorgement only in db/db mice. The immune cell population was basically different between Naïve mice and db/db mice. Ephedrine treatment has reduced splenocytes in db/db mice and changed statistically significantly Mac3+ cell population. The author also referred to the other markers that tend to be changed. Then the authors tested to NK activity, macrophage function, and mitogen-induced lymphocyte proliferation. The authors also checked the cytokine secretion and gene profiles. The stream of the experiments is acceptable.

The discussion section presented only a few references. Ex) Lines 303-306; the authors discuss about cxcl4 and cxcl7 without any references. The alterations of the expression patterns of IL-12 and -10, shown in Fig.6 should be referred to in the Discussion. Especially, the Discussion section lacks the argument about beta-androgenic receptor agonist properties of ephedrine.

Like,

https://www.sciencedirect.com/science/article/pii/S0091674900901830

https://www.sciencedirect.com/science/article/pii/S0889159118305555

and/or other related articles.

[Figures and Tables]

Figure quality; Great, but be careful for the resolution

Readability; Characters in the Fig 1 and 2 should be bigger. Throughout the Figures, Through the manuscript, the abbreviation of ephedrine has fluctuations eph and ephe. Please unify. In addition, please show the full name without abbreviation in the Figure Legend

Comments for each figures;

Figure 5

1) The authors should also present the significance of ephedrine-treated group vs non-treated group in C57BL/6 mice.

[Data presenting]

If the author used flow-cytometry analyses, it is better to show the representing scatter plot.

 [References]

The references used are few, partly because of the lack of refs in the Discussion section. Generally, the number of references will reach 30 refs in such a scale of the manuscript. Some references have doubts about the proper citation. Ex), Line 271 says “Our results showed that … [13]” However, the authors of Ref 13 are different from the authors of this manuscript. Line 57 also says “Our previous data reveal significantly elevated … [12]”, but the authors of Ref 12 are different from the authors of this manuscript. The authors should check the correspondence of references totally.

[Minor points]

Line 40: Ma Whang may be Ma Hwang.

Line 83-84: Canavalia ensiformis and Escherichia coli should be italic.

Line 116, 129, 140, and others: it should be superscript

Comments on the Quality of English Language

There are some typos but quality is average.

Author Response

Thank you very much for taking the time to review this manuscript and

Reviewer 2 Report

Comments and Suggestions for Authors

Ephedrine is one of the most powerful fat reduction products. That knowledge is not new. Additionally, ephedrine has been banned by major organisations such as the International Olympic Committee, the World Anti-Doping Agency. So, the question is if the manuscript has got merit? The answer is: yes. The present manuscript demonstrates the ephedrine-induced alteration in the systemic immunity in the obese diabetic mouse model and I agree that those data may expand our understanding of obesity and related diseases in terms of the mechanism of induction or development of therapeutics. Only few remarks and I would recommend this paper for publication.

Please explain very carefully: normally, the effects of ephedrine associated with loss of weight can be seen after continuous application. Here, you treated mice with a single dosage and the animals were euthanized after 24 hours. I am working in the field of nutritional studies with rodents and quite often I am involved in pharmacological experiments but I am very surprised how the histological changes can be so huge after a single dosage? Maybe the applied dosages were too high> I read the explanation in the text but the observed changes were incredible. Do you plan an experiment with long-term treatments?

The experimental schema is very appropriate. The language understandable. The analytical methods OK.

Figure 7: why you omit the ephe3mg treatment?

Author Response

(The authors gave the same response as above.)

Round 2

Reviewer 1 Report

Comments and Suggestions for Authors

The authors responded to my suggestion and corrected it properly. However, the authors did not discuss the relation between the immunomodulation activity and the beta-androgenic receptor agonist activity of ephedrine. In other words, is the immunomodulation activity blamed for the ephedrine beta-androgenic receptor agonist activity or not? Please explain or discuss it in the discussion section.

Author Response

(The authors gave the same response as above.)

Round 3

Reviewer 1 Report

Comments and Suggestions for Authors

The authors responded to my question as 

"In this study, the correlation between the already known adrenergic receptor-related activity and systemic immune modulation was not investigated. Further studies exploring the relationship between these two mechanisms of action would be valuable."

If the sentences are included in the revised manuscript, the manuscript will be ready for publication.

Author Response

Thank you very much for taking the time to review this manuscript and

OR

Please see revised manuscript in the re-submitted files; lines 310-312
